# A Method for Assessing Threats to the Economic Security of a Region: A Case Study of Public Procurement in Russia

**Valentina Kravchenko** * , **Tatiana Kudryavtseva** and **Yuriy Kuporov**

Peter the Great St. Petersburg Polytechnic University, Institute of Industrial Management, Economics and Trade, Graduate School of Industrial Economics, 195251 St. Petersburg, Russia; kudryavtseva_tyu@spbstu.ru (T.K.); kuporov_yuyu@spbstu.ru (Y.K.)

* Correspondence: kravchenko_vv@spbstu.ru; Tel.: +7-911-285-14-49

**Abstract:** The issue of economic security is becoming an increasingly urgent one. The purpose of this article is to develop a method for assessing threats to the economic security of the Russian region. This method is based on step-by-step actions: first of all, choosing an element of the region's economic security system and collecting its descriptive indicators; then grouping indicators by admittance-process-result categories and building hypotheses about their influence; testing hypotheses using a statistical package and choosing the most significant connections, which can pose a threat to the economic security of the region; thereafter ranking regions by the level of threats and developing further recommendations. The importance of this method is that with the help of grouping regions (territory of a country) based on proposed method, it is possible to develop individual economic security monitoring tools. As a result, the efficiency of that country's region can be higher. In this work, the proposed method was tested in the framework of public procurement in Russia. A total of 14 indicators of procurement activity were collected for each region of the Russian Federation for the period from 2014 to 2018. Regression models were built on the basis of the grouped indicators. Ordinary Least Squares (OLS) Estimation was used. As a result of pairwise regression models analysis, we have defined four significant relationships between public procurement indicators. There are positive connections between contracts that require collateral and the percentage of tolerances, between the number of bidders and the number of regular suppliers, between the number of bidders and the average price drop, and between the number of purchases made from a single supplier and the number of contracts concluded without reduction. It was determined that the greatest risks for the system were associated with the connection between competition and budget savings. It was proposed to rank analyzed regions into four groups: ineffective government procurement, effective government procurement, and government procurement that threatens the system of economic security of the region, that is, high competition with low savings and low competition with high savings. Based on these groups, individual economic security monitoring tools can be developed for each region.

**Keywords:** economic security of the region; economic security system; public procurement; regression analysis; regions of Russia

## 1. Introduction

Economic security is a national security issue. The economic security of any state is affected by the economic security of all its states, regions, and districts. The economic security of a state defines the stability and progressive development of the economy of this territory (Kahler 2004; Gryshova et al. 2020; Feofilova 2013; Gutman et al. 2018; Uspenskij et al. 2019). The economic security of a region also determines the current state, conditions and factors that directly contribute to or oppose this development (Hacker et al. 2013; Feofilova et al. 2018). Economic security is seen as an endogenous factor of regional development and of the ability of the region as a system to achieve targets with

the efficient use of resources and to maintain its inherent attribution characteristics under the conditions of instability and uncertainty in the external and internal environment, that is, the presence of some threats and risks (Husainova et al. 2019). It follows that for effective control of this system, it is necessary to clearly identify the internal foci of risks and threats.

There are different approaches to the meaning of an economic security system and its elements. Due to efforts by the Financial Action Task Force (FATF), money laundering, terrorist financing, and funding for weapons of mass destruction are prohibited in an economic security system. Scientific ideas have defined an economic security system as an effective independent regional system that can develop independently (Rossinskaya and Bugaeva 2010; Vik 2012). Other authors include threats and the need for measures to protect and counteract these threats under this definition (Voronin 2001; Litvinenko and Samoilova 2017). Some authors combine these approaches into a state system, which protects the economy from the effects of negative impacts (Feofilova 2014). One author emphasized the importance of spheres of an economy, which directly determine the "course of the reproductive process in the region" (Sizov 2005). By aggregating all the concepts, we obtain the socio-economic system of the region, within which various elements (Element 1-n) interact based on the principles of efficiency, competitiveness, stability, and sustainability, as well as on the ability to manage and prevent internal and external threats to its development (Figure 1).

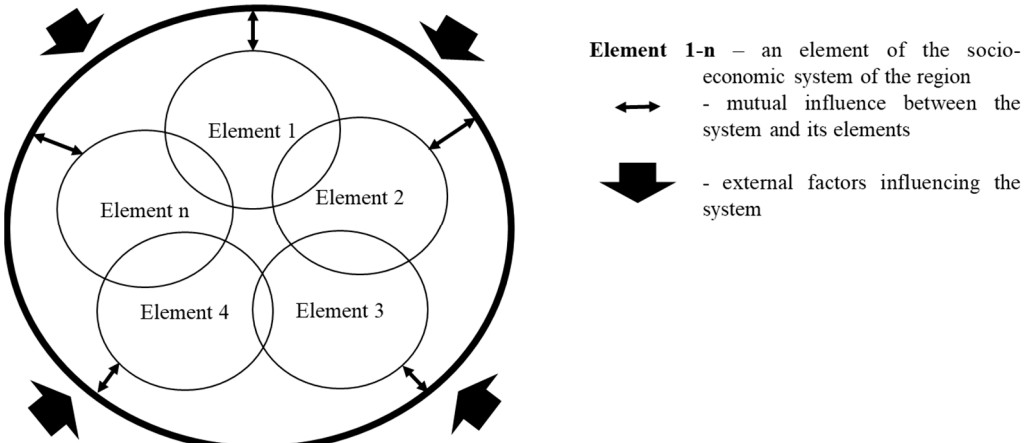

**Figure 1.** System of economic security of a region.

Figure 1 schematically shows that the system of economic security of a region consists of a set of interconnected elements. Each element has regulatory and legal acts that govern subjects of management, objects of management, the possibility of negative impacts, and consequences of their impact on the system, which are a group of indicators and indicators that characterize the system (Feofilova 2014). In the system, the elements interact. Its environments both positively and negatively affect the general environment of economic security of the region. Simultaneously, the system of economic security, which is exposed to external factors, has both a positive impact on its elements and a negative impact on its elements.

From the authors' point of view, it is important to identify threats that arise inside the system. For achieving this idea, it is important to understand the locations of the threats, that is, we need to know the elements of the system that can threaten it. Therefore, the purpose of this paper is to create a universal method for assessing threats to the economic security of the region. This method examines system elements that can potentially cause threats.

## 2. Results

This work focuses on public procurement, which is one of the most important elements of the system of economic security of a region (Grandia and Kruyen 2020; Jacobson 2000). The process of the developed method involves assessing threats to the economic security of

the Russian regions through the framework of public procurement. The stages and details of the method are presented in the 4th part of this article, Materials and Method.

Stage 1. In this study, public procurement was selected as an element of the region's economic security system that can influence it.

Stage 2. The collection of descriptive indicators on public procurement in the regions occurred via the Tenderplan analytical platform. Tenderplan (https://tenderplan.ru/) is a system for searching and analyzing tenders; it includes data on purchases, suppliers, and customers.

Descriptive indicators of public procurement on the Tenderplan analytical platform include 14 indicators: (1) the number of tenders held in the region in one year; (2) the cost equivalent of all tenders held in the region in one year; (3) the value of all completed tenders in the region in one year; (4) the percentage of purchases made from small- and medium-sized businesses in the region for one year; (5) the average ratio of the declared participants in the tender to those who actually participate for each purchase in the region for one year; (6) the number of suppliers that constantly participate in the tenders in the region for one year; (7) the average price of a concluded contract in the region for one year; (8) the average price drop among all contracts in the region for the year; (9) the percentage of tenders that require collateral among all tenders in the region in one year; (10) the number of contracts that are concluded without reducing the initial (maximum) price in the region for bidders in one tender among all tenders in the region for one year; (11) the number of participants in all tenders in the region for one year; (12) the average number of participants in one tender among all tenders in the region for one year; (13) the number of customers who constantly place orders in the region in one year; (14) the number of contracts that are concluded with a single supplier without participation in the competitive procedure in the region for one year; and (15) the percentage of contracts for which post-contract complaints were made in the region in one year. Numbering of indicators from 1 to 15 is used in this work.

Stage 3. These indicators for public procurement in the regions can be grouped according to the admittance-process-result distribution. The detailed logic of distribution is shown in the Material and Methods part of this paper. The indicators of admittance to the participation in tenders are responsible for the "admittance": indicators (5), (9), and (13). The "process" are the indicators that directly characterize the competitive procedures: (6), (11), and (12); the "result" comprises the final indicators of the procurement procedures: (4), (7), (8), (10), (14), and (15).

Stage 4. Based on the groups obtained, their influence on each other was revealed: the admittance group affects the process and the results, and the process group affects the results group. Hypotheses were formed to test them.

Stage 5. For statistical processing of the obtained data and verification of the formed hypotheses, we used the Stata software product, which provides data processing, visualization, statistics, and automatic reporting. This product is convenient and simple for the regression analysis, which allowed us to determine the connections among the grouped indicators (4–15). One connection acts as a dependent variable, and the other connection acts as an independent variable. The independent variables (4–15) are horizontally displayed in Table 1, whereas the dependent variables (4–15) are vertically displayed in Table 1.

The summary table (Table 1) shows the direction of the changes in the related indicators (4–15), provided that the indicator is significant. The significance of the indicator was determined based on the Student's t-test: ("↑↑")—indicators have a co-directional change, that is, with an increase (decrease) in the influencing indicator, the dependent variable increases (decreases); ("↑↓")—indicators have a multidirectional change, that is, with an increase (decrease) in the influencing indicator, the dependent variable decreases (increases). If the value of the t-criterion is greater than the significance level, then the null hypothesis about the insignificance of the regression coefficient is accepted, that is, there is no connection among the analyzed indicators "-". The influence of the variable on itself is not checked "X".

**Table 1.** Results of regression analysis: direction of changes in related indicators.

| y \ x | 4 | 5 | 6 | 7 | 8 | 9 | 10 | 11 | 12 | 13 | 14 | 15 |
|---|---|---|---|---|---|---|---|---|---|---|---|---|
| 4 | X | ↑↑ | ↑↑ | - | ↑↑ | ↑↑ | ↑↓ | - | ↑↑ | - | - | - |
| 5 | ↑↑ | X | - | ↑↑ | ↑↑ | ↑↑ | ↑↑ | - | ↑↑ | - | - | - |
| 6 | ↑↑ | - | X | - | ↑↑ | ↑↓ | ↑↑ | ↑↑ | ↑↑ | ↑↑ | ↑↑ | - |
| 7 | - | ↑↑ | - | X | ↑↑ | ↑↑ | ↑↓ | - | - | - | ↑↑ | - |
| 8 | ↑↑ | ↑↑ | ↑↑ | ↑↑ | X | ↑↑ | ↑↓ | ↑↑ | ↑↑ | - | ↑↓ | - |
| 9 | ↑↑ | ↑↑ | ↑↓ | ↑↑ | ↑↑ | X | ↑↓ | ↑↓ | ↑↑ | - | ↑↓ | - |
| 10 | - | ↑↑ | ↑↑ | ↑↓ | ↑↓ | ↑↓ | X | ↑↑ | ↑↓ | ↑↑ | ↑↑ | - |
| 11 | - | - | ↑↑ | - | ↑↑ | ↑↓ | ↑↑ | X | ↑↑ | ↑↑ | ↑↑ | - |
| 12 | ↑↑ | ↑↑ | ↑↑ | - | ↑↑ | ↑↑ | ↑↓ | ↑↑ | X | ↑↑ | ↑↓ | - |
| 13 | - | - | ↑↑ | - | - | - | ↑↑ | ↑↑ | ↑↑ | X | ↑↑ | ↑↓ |
| 14 | - | - | ↑↑ | ↑↑ | ↑↓ | ↑↓ | ↑↑ | ↑↑ | ↑↓ | ↑↑ | X | - |
| 15 | - | - | - | - | - | - | - | - | - | ↑↓ | - | X |

Stage 6. We considered the most significant connections revealed in the process of statistical data processing. The column Model shows the number of dependent and independent variables. β-coefficient shows how much we expected the dependent variable to change by given a change in the independent variable. R-square shows the amount of variance of dependent variable explained by independent variable. Regression standard error (SE) measures the squared error per degree of freedom of the model; the lower the SE, the better the model. The *p*-value was used to show that each coefficient is different from 0.

The most significant connections described in the Table 2 can be represented and explained in the way of pair regression equations.

$$y = 45.77 + 0.56x, \tag{1}$$

where x is the number of applications that require collateral, and y is the percentage of tolerances.

**Table 2.** Main indicators of regression analysis.

| Number of Equation | Model | SE | β | *p*-Value | R-Squared |
|---|---|---|---|---|---|
| 1 | (5) | | | | 0.6341 |
| | (9) | 0.0197 | 0.5589 | 0.00 | |
| 2 | (6) | | | | 0.9437 |
| | (11) | 0.0048 | 0.4265 | 0.00 | |
| 3 | (7) | | | | 0.5367 |
| | (12) | 0.2649 | 6.1278 | 0.00 | |
| 4 | (10) | | | | 0.5527 |
| | (14) | 0.0080 | 0.1921 | 0.00 | |

In the presented model, on average, an increase in the percentage of the number of applications that require collateral by one point will cause an increase in the percentage of tolerances by 0.56 points:

$$y = 148.95 + 0.43x, \tag{2}$$

where x is the number of participants in procurement, and y is the number of regular suppliers.

There is a positive connection among indicators, such as the number of regular suppliers and the number of participants. On average, if the indicator Number of participants increases by one unit, then the indicator Number of regular suppliers increases by 0.43 points.

$$y = 3.5 + 6.12x, \tag{3}$$

where x is the average number of bidders, y is the average price drop.

There is a positive connection among the indicators. If the average number of bidders increases by one, then the average price drop will increase by 6.12. Note that variable x explains 53% of the changes in variable y:

$$y = -186.25 + 0.19x, \tag{4}$$

where x is the number of contracts with a single supplier, and y are contracts without reduction.

There is a positive connection among the presented indicators. On average, an increase in contracts with a single supplier by one point will entail an increase in the number of contracts without a decrease of 0.19. Note that the variable x explains 55% of the changes in the variable y.

As a result of testing paired regression models, the main laws of the market were confirmed, according to which the instrument of state regulation, government procurement works, and the relationships that potentially threaten the system of economic security of the region were identified.

First, these indicators of admittance as the average ratio of the declared participants in the tender to those that actually participate for each purchase in the region for one year and the applications that require security are interconnected. Second, contracts with non-price reductions and contracts with a single supplier are mutually influencing, that is, in regions with a low level of competition, more contracts are concluded without reductions and contracts with a single supplier. Third, the average number of bidders affects the average price drop. Due to healthy competition, producers of goods and services attempt to focus on improving the satisfaction of society's needs. This rule also applies to public procurement (Drozdov and Hlystova 2019). According to the authors, the lack of competition in public procurement entails monopolization of the market, which causes an increase in producer costs and a decrease in society's satisfaction from the benefits received. There is no return on costs caused by the results and compliance with the accepted evaluation criteria. These consequences can lead to an unstable process of economic development and socio-economic stability for a society, which directly poses a threat to the economic security of not only regions but also the country as a whole (Chupanova and Gasanalieva 2016; Drozdov and Hlystova 2019).

Moreover, there is a co-directional change among indicators, such as the percentage of purchases from small- and medium-sized enterprises in the region and the average price drop in the region; there is also a co-directional change for the average contract price and the number of contracts with a single supplier. Simultaneously, multidirectional changes have connections such as: the percentage of purchases from small- and medium-sized enterprises and the number of contracts without reduction; the number of contracts without reduction and the average contract price; the number of contracts without reduction and the average price drop; as well as the number of contracts with a single supplier and the average decline prices in the region during the procurement procedure.

Stage 7. The literature confirms the authors' opinions that corruption is widespread in the field of public procurement (Wang 2020; (Evlanova and Pavlovskaya 2015; Chupanova and Gasanalieva 2016; Feofilova and Yarilova 2019). Corrupt behaviors of officials entail unlawful admission to participation in the competition and/or determination of the winner of the supplier, whose application is officially subject to rejection (Huang and Xia 2019; Peshkov and Klimanov 2016). This kind of behavior can contribute to the participation in the tenders of affiliated suppliers and prior agreement between the participants on the declared price of the contract (Dastidar and Mukherjee 2014; Owusu et al. 2019; Lavrov and Lapin 2017). These examples show that some documented competition is legitimate and real.

Based on this information, it was proposed to assign each region to one group of four regions with respect to the connection between the average number of bidders per purchase and the average price drop (Table 3).

(1) Group 1—ineffective public procurements. Regions with low competition, which yield low price reductions;
(2) Group 2—a threat to economic security. Regions with high levels of competition but suspiciously low price reductions;
(3) Group 3—a threat to economic security. Regions with suspiciously high price levels with minimal competition;
(4) Group 4—effective public procurements. Regions with a favorable atmosphere for the development of the public procurement system, that is, a high level of high-price reductions.

**Table 3.** Groups of regions regarding the connection between competition and economy.

| Number of Bidders in One Purchase in the Region | Decrease in the Initial Price for One Purchase in the Region | |
| --- | --- | --- |
| | **Below Average for Russia** | **Above Average for Russia** |
| Above average for Russia | (3) | (4) |
| Below average for Russia | (1) | (2) |

The average values of indicators, average number of bidders in one procurement ($N_{av}$) and average price drop in one procurement (E) were the quartile boundaries for assessing the procurement activity of the regions of the Russian Federation (RF) (Table 4).

**Table 4.** Borders of proposed groups.

| | **Average Price Drop, %** | **Average Number of Bidders, pcs.** |
| --- | --- | --- |
| 2014 | 13.32619 | 2.097619 |
| 2015 | 17.72381 | 2.478571 |
| 2016 | 18.63537 | 2.463415 |
| 2017 | 20.66279 | 2.630233 |
| 2018 | 19.78488 | 2.594186 |

## 3. Discussion

We considered the development trend in the regions of the RF in terms of the relationship between competition and economy in the period from 2014 to 2018 (Table 5).

**Table 5.** Number of regions in the group, 2014–2018.

| | **Group 1** | **Group 2** | **Group 3** | **Group 4** |
| --- | --- | --- | --- | --- |
| 2014 | 10 | 4 | 12 | 16 |
| 2015 | 28 | 14 | 8 | 34 |
| 2016 | 28 | 10 | 9 | 35 |
| 2017 | 24 | 10 | 13 | 39 |
| 2018 | 24 | 13 | 11 | 38 |

Note: there is no information for 2014 for 44 regions: Amur Region, Jewish Autonomous Region, Kabardino-Balkaria, Kalmykia, Karachay-Cherkessia, Karelia, Kirov Region, Komi, Kostroma Region, Crimea, Kurgan Region, Leningrad Region, Lipetsk Region, Magadan Region, Mari El, Mordovia, Moscow, Moscow Region, Nizhny Novgorod Region, Novgorod Region, Novosibirsk Region, Omsk Region, Orenburg Region, Penza Region, Perm Region, Primorsky Region, Pskov Region, Samara Region, St. Petersburg, Saratov Region, Smolensk Region, Stavropol Region, Tambov Region, Tatarstan, Tver Region, Tomsk region, Tula region, Tyva, Tyumen region, Ulyanovsk region, Khabarovsk region, Chelyabinsk region, Yamalo-Nenets Autonomous District, Yaroslavl region; in 2015—Ryazan region, Sakhalin region; in 2016—Vologda region, Sevastopol, Smolensk region, Tyumen region.

In 2014, 10 regions fell into the first group of regions with a low level of competition and a low drop in prices: Altai, Voronezh Region, Dagestan, Krasnoyarsk Territory, Rostov Region, Sakhalin Region, North Ossetia—Alania, Chechnya, Chukotka Autonomous Okrug, and Yakutia. In 2018, 24 regions fell into the same category: Adygea, Amur Region, Arkhangelsk Region, Belgorod Region, Buryatia, Vologda Region, Dagestan, Ivanovo Region, Ingushetia, Kalmykia, Karachay-Cherkessia, Kostroma Region, Krasnodar Territory,

Magadan Region, Nizhny Novgorod region, Primorsky Territory, Pskov Region, Sakhalin Region, North Ossetia—Alania, Tula Region, Khabarovsk Territory, Chechnya, Chukotka Autonomous Okrug, and Yakutia. The second group included 12 regions in 2014 and 16 regions in 2018. The third group included 4 regions in 2014 and 11 regions in 2018. The fourth group included 16 regions in 2014 and 38 regions in 2018. The numerical distribution of the regions by groups from 2014 to 2018 is presented in Table 5 and Figure 2.

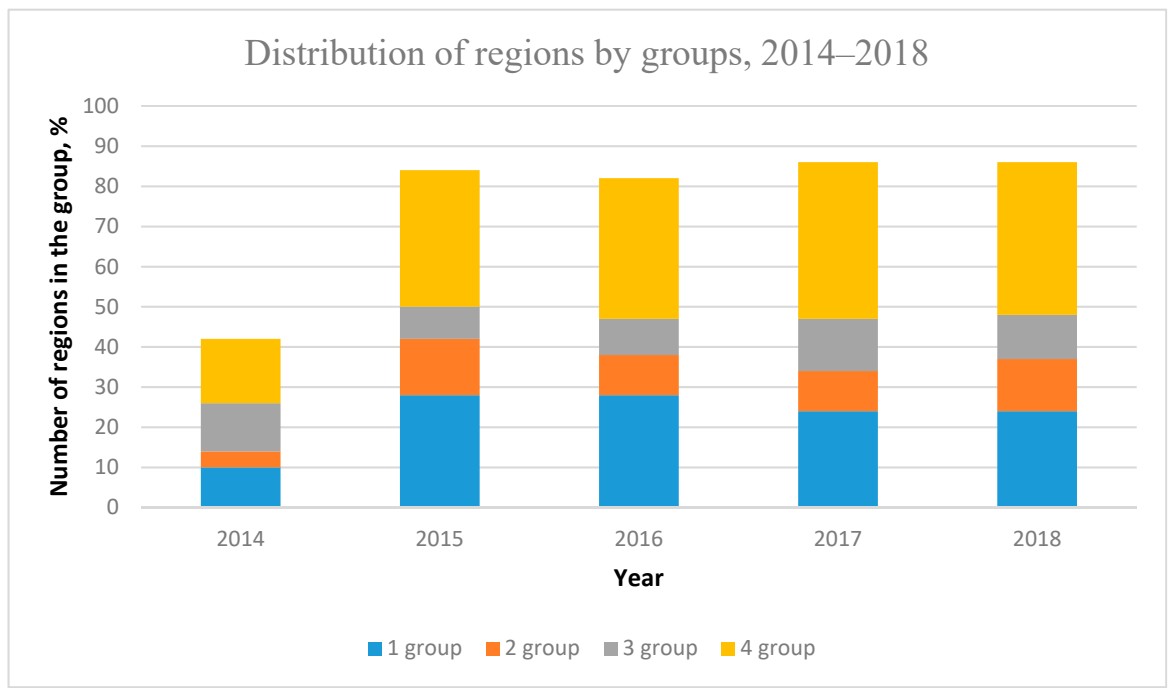

**Figure 2.** Distribution of regions by groups, 2014–2018.

A characteristic dynamic over the course of the five years is an increase in the number of regions with an indicator of competition that is above average with high budget savings. This finding may indicate the success of the RF contract system. However, in more than half of the regions, the connection is unsatisfactory, that is, there is a threat to the economic security of the region or the system of public procurement is ineffective. In this context, it makes sense to pay attention to the work of the mechanism of state orders in these regions, since this may be associated with not only its specifics but also the external and internal violations in the organization of the public procurement procedure.

At present, in accordance with the "Instruction on the implementation of procurement control activities in the Federal Treasury", the drafting of the audit plan is drawn up on the basis of the sectoral and risk-oriented approaches. It is proposed to additionally use the developed methodology for assessing threats to the system of economic security of the region from the side of public procurement.

The combination of three approaches in drawing up a plan of inspections of the Federal Treasury will make it possible to form a more complete list of objects, which will prevent the spread of threats to the economic security of the region.

### 4. Materials and Methods

In general, the method for assessing threats to the economic security of the region from its elements can be a sequence of interrelated actions that lead to ranking regions regarding threats and the resulting measures to counter their spread. These measures should be considered by the state authorities. The method is shown in Figure 3.

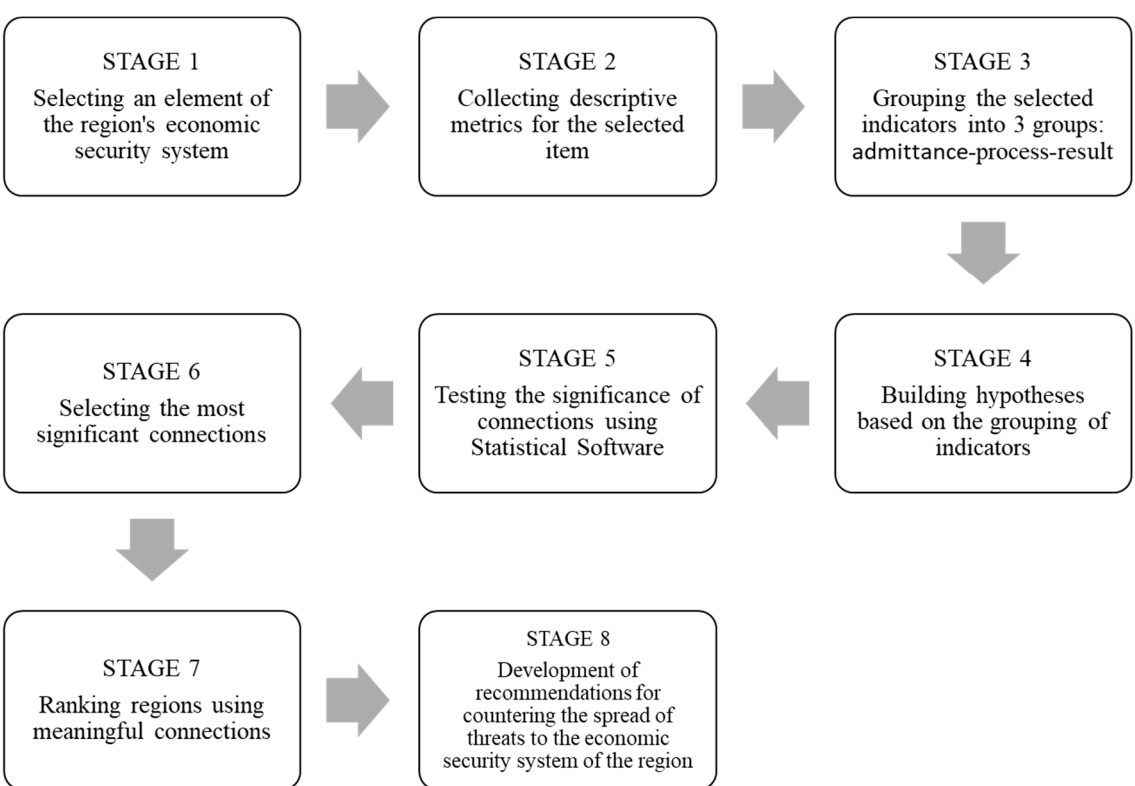

**Figure 3.** Stages of the method for assessing threats to the economic security of the region.

It is necessary to select an element of the region's economic security system and collect indicators that describe its activities. A separate justification is required for the third stage of the proposed method, grouping the selected indicators into three groups: admittance-process-result, which is similar to the well-known chain of input–throughput–output (de Bruyne 2005). The public sector is an essential part of a region's economic security system. For this reason, the grouping of indicators was based on an analogy with the assessment of the efficiency of the public sector proposed by the American researcher Hans de Bruyne, which has been proven to require a combined system, that is, it is advisable to concurrently evaluate the product and process (Figure 4).

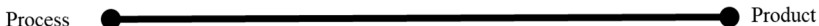

**Figure 4.** Playing field for evaluating the effectiveness of the public sector.

This work purposefully adds a missing element of this chain: the preparatory stage, which is a kind of resource interpretation (Figure 5). In addition, there is a replacement of "product" with "result", since the elements of the region's economic security system still represent interconnected subsystems, in which the activities can be determined by results instead of products. The combination of these three components determines the causes of threats.

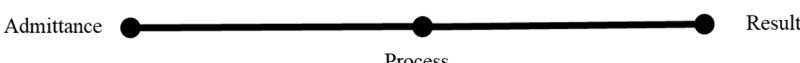

**Figure 5.** Playing field for evaluating the effectiveness of an element of the region's economic security system: admittance–process–result.

In addition, grouping a certain element into three related groups (admittance–pro–cess–result) takes into account the dynamics of the development of both the external security environment of the region and the internal environment of the element that is being considered.

Hypotheses are based on the connections within and outside the formed groups. Using statistical analysis, significant connections can be determined, which can be subsequently employed as key steps for ranking regions by the level of threats to the economic security of the region. Based on the connections obtained and the results of the analysis of the regions, it is possible to develop recommendations to prevent the spread of threats to the entire system of economic security of a region.

## 5. Conclusions

A method for assessing threats to the system of economic security of the region from system's elements was developed. The method consists of eight sequential stages, which can be applied to every element of the system of economic security of the region or territory of every country.

Approbation of the method in the framework of public procurement in Russian regions revealed that the non-standard ratio of competition indicators to economy indicators entails a threat to the economic security of the region. Based on this connection, four groups of regions were proposed. Regions of the RF were separated into groups. From the analysis of the dynamics of the regions of the RF, the groups were divided into several categories.

The first category is characterized by a stable fall into the quartile of ineffective public procurement. During the analyzed period, the competition indicators were below average for Russia. In addition, the average price drop did not exceed the national average. This category includes Dagestan, North Ossetia—Alania, and Chechnya. The second category is characterized by a stable positive situation, that is, over the course of five years, competition and the fall in prices were above average for Russia. These regions include Altai Territory, Arkhangelsk Region, Astrakhan Region, Vladimir Region, Crimea, Kaluga Region, and Mordovia. The third category includes the unstable regions. These include regions with a gradual improvement in public procurement, such as the Magadan region and Moscow region, and one year of a worsening public procurement situation, such as the Novgorod region and Orenburg region.

Based on the proposed method, a tool for monitoring the degree of economic security in the region for the field of public procurement can be created, which enables the tracking of a potential threat to the entire system of economic security during tendering procedures using digital technologies. This method can be scaled at the level of cities in any country, which renders its use to be relevant in a country that is concerned with its economic security.

**Author Contributions:** Conceptualization, V.K. and T.K.; methodology, V.K.; formal analysis, V.K.; investigation, V.K., T.K., and Y.K.; resources, V.K.; data curation, V.K.; writing—original draft preparation, V.K. and T.K.; writing—review and editing, T.K.; visualization, V.K.; supervision, T.K.; project administration, T.K. All authors have read and agreed to the published version of the manuscript.

**Funding:** Ministry of Science and Higher Education of the Russian Federation: The Academic Excellence Project 5-100 proposed by Peter the Great St. Petersburg Polytechnic University.

**Institutional Review Board Statement:** Not applicable.

**Informed Consent Statement:** Not applicable.

**Data Availability Statement:** The data presented in this study are available on request from the corresponding author.

**Acknowledgments:** This research work was supported by the Academic Excellence Project 5-100 proposed by Peter the Great St. Petersburg Polytechnic University.

**Conflicts of Interest:** The authors declare no conflict of interest.

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
