# Peer review of "A Method for Assessing Threats to the Economic Security of a Region: A Case Study of Public Procurement in Russia"

_risks, doi:10.3390/risks9010010_

Round 1

Reviewer 1 Report

The article is interesting and generally, it deserves to be published with some revisions that are suggested below:

Data: Can you state how many quantity for the relating data that you collected from the Tenderplan analytical platform in stage 2? and how many valuable data that you used after cleaning data?

Please let us know what the indicators which are the independent variable and dependent variable in stage 5?

Author Response

Thank you for your comments.

  1. We have added that Tenderplan offers 14 indicators and just 12 of them are used in the research.

"Descriptive indicators of public procurement made under 44 federal laws on the Tenderplan analytical platform include 14 indicators: the number of tenders held in the region in one year; the cost equivalent of all tenders held in the region for one year; the value of all completed tenders in the region in one year; (1) the percentage of purchases made from small and medium-sized businesses in the region for one year; (2) average ratio of the declared participants in the tender to those actually participating for each purchase in the region for one year; (3) the number of suppliers constantly participating in tenders in the region for one year; (4) the average price of a concluded contract in the region for one year; (5) average price drop among all contracts in the region for the year; (6) percentage of tenders requiring collateral among all tenders in the region in one year; (7) the number of contracts concluded without reducing the initial (maximum) price in the region for bidders in one tender among all tenders in the region for one year; (10) the number of customers who constantly place orders in the region in one year; (11) the number of contracts concluded with a single supplier without participation in the competitive procedure in the region for one year; (12) percentage of contracts for which post-contract complaints were made in the region in one year. Indicators number (1)-(12) are used In this research."

2. We have added the sentence "There are independent variables horizontally in the resulting Table 1, dependent variables – vertically." for understanding which variable is dependent or independent in single connection.

Reviewer 2 Report

It's methodological paper, so the it has to be written in the way, that other scientiscts could repeat the method. From this one there is no such a posibility. There is a case study of Russian regions, but authors didn't provide information if and how the method could be repeated in other countries. 

  1. Introduction:

- After reading the paper it could be sayed that only russian authors has studied the paper's topis, but it's not true. In introduction more international authors should be added. This part is too short, and more references should be added in this part.

- The purpose of the paper isn't clear and has to be strenghtened and wider written.

2. Results.

- Line 88, Stage 3. Explain more, why the indicators were groupped to mentioned 3 groups.

- I couldn't find any calculations and mathematic formulas how the results were calculated. Is has to be added.

- Table 3. The regions aggregated to groups should be written.

3. Materials and methods.

- line 223  - citation in english

4. Conclusions

- Risks is an interantional journal, but this paper doesn't provide any international implications or conclusions that coul adapted in other countries. In this case I don't think it could be interested for international audience. 

Author Response

Thank you for your comments.

There is detailed description of the method in the 4th part of the article (Materials and Method), especially Figure 3. The use of this method is presented a case study of Russia, but using the method (Figure 3) it could be repeated in every country.

Indroduction.

Results.

We've added the sentence about logic of distribution "Detailed logic of distribution is shown in the part of Material and Mythology of this paper. ", as there we have described it fully.

"we use the Stata software product" for making regression analyse, the results are shown in the Table 1 and 2.

Table 3. The regions aggregated to groups should be written. - The regions are shown further in the text. Table 3 is used to analyze what group of region it can relate to. We have changed the sentense for better understanding "Based on the foregoing, it is proposed to assign each region to one group of four with respect to the connection between the average number of bidders per purchase and the average price drop (Table 3)."

Conclusion.

"Also this method can be scaled at the level of cities in any country what makes it relevant for using in a country worried about its economic security."

Reviewer 3 Report

Highlight changes in yellow in a next revision, please. No track changes.

Consider comments in the entire text.

Revise title: “The” means there is no possible other…

Also, to start by the is not enlightening…

The title s important, reframe it:

 The method for assessing threats to the economic 2 security system of the region in the framework of 3 public procurement

See that the language is not clear and then it will reflect on the entire text...

Abstract…

What region? Economic security at what level or context?

“more topical”?

“Currently, the issue of economic security of the region is becoming more topical.”

A method, not the method… “ study is to develop a method”, compare with title…

Where? See that the title does not mention Russia… “regional economic”

I do not understand the sentence: “to the regional economic security 12 system from its elements.” What elements?!

Revise language, determined means something else:

“It was determined that the greatest risks for the system was in connection between competition and 23 budget savings.”

End with practical implications to highlight the importance of this text…

Keywords cannot be presented like this: “Economic security of the region;”

Maybe “Russia Economic security”

Revise the entire language mentioning region… in the complete manuscript…

Correct, have you double checked? “state procprocurement”

As seen, the English must be entirely proofread… It is not correct, just as example:

“In modern conditions of 36 digitalization progress can have as supporting influence to the system economic security of a region as 37 the potential threats”

This says nothing… Elements from 1 to n, no detail at all, see that Figures must be original and captions complete self-explanatory, which is not the case (the region “isse” again…):

Figure 1. – The system of economic security of the region.

Clarify…

“This figure schematically shows that the system of economic security of the region consists of a set of 59 interconnected elements with its own internal environment.”

After Introduction, results follow.

I acknowledge  authors are strictly following the template: https://www.mdpi.com/files/word-templates/risks-template.dot, here: https://www.mdpi.com/journal/risks/instructions.

Most of the Reviewers will be misled, because then the coherence is not there when authors state: “2. Results 65

The proposed method was tested on the assessment of threats to the economic security of the 66 region from public procurement by the following steps”

It is not a question of respecting the template, but also to adequate the text to the order indicated in the temple. The method was not yet presented…

Are theses results or methodology? See:

“Stage 1. In this study, public procurement was selected as an element of the region's economic 68 security system that can influence it.”

Authors are not clear, an international reader needs proper references for all similar cases…

“44 federal laws on the Tenderplan 74 analytical platform”

Revise all italics to variables: “Student's 102 t-test:”

Revise captions to be completely self-explanatory: “Table 1. – Results of regression analysis.”

Revise all headings in captions and add notes at the end of the tables, because they are available separately from the text:

y]x

”ed

There cannot be similar captions, not possible to different tables:

Table 2. - Results from regression analyses.”

And check English too

Again, italics, headings , notes, to be checked…

Clarify statements and hypothesis…

Which are presented later on? Methodology…

The structure of the text needs complete revision then…

“H1: The indicator percentage of applications requiring collateral to participate in the tender 114 affects the percentage of admissions.”

All equations must be properly introduced by number previously to being presented, as usual… and italics and units must be checked, after equations

Address the originality of equations to, no references means complete originality.

References are not presented after dots…

“but also the country as a whole. 152 (Chupaniva, 2016; Drozdov, 2019) 153

Fourthly, there is a co-directional”

Do not use the “listing” style: “Fourthly, there is”

One author then?! Possessive case… check English…

“Stage 7. The studied literature confirms the author's opinion”

What regions, what groups?

Table 3. - Distribution of regions by groups.”

Groups defined after the table? Add notes, there are much better ways to present nad clarity the data…

“Group 1 - ineffective state procurements. Regions with low competition, resulting in low price 174 reductions; 175

Group 2 - a threat to economic security. Regions with high levels of competition but suspiciously 176 low price reductions; 177

Group 3 - a threat to economic security. Regions with suspiciously high price levels with little 178 competition; 179

Group 4 - effective state procurements. Regions with a favorable atmosphere for the 180 development of the public procurement system, that is, a high level of high price reductions.”

Units are presented inside () and there must be headings to all:

Table 4. - Borders of quartiles for assessing the procurement activities of the regions of the Russian 186 Federation.”

Language must be revised…

3. Discussion 188

Consider the development trend of the regions of the Russian Federation in terms of the relationship 189 between competition and economy in the period from 2014 to 2018 (Table 5).”

And again, the discussion contains tables and figures, is it a discussion or results sections, either authors join or reframe…

A discussion should justify/support/debate the results and include references if necessary, which is not the case

All axis must have legends and units etc, check all, the first authors seems not to be used to the publish process but others are…

Not “1 group” but “group 1”, etc…

And again, the necessary clarification: “Figure 2. – Distribution of regions by groups, 2014-2018.”

 Believe authors must dedicate a significant further effort in revising the entire language, namely considering the followed structure in the manuscript…

4. Materials and Methods 213

In general, a method for assessing threats to the economic security of a region from its elements 214 can be a sequence of interrelated actions leading to ranking regions regarding threats and the 215 resulting measures to counter their spread. These measures are supposed to be considered by the 216 state authorities. The method is shown in the Figure 3.”

Then a method or the method again… or “region”…

Figure 3 – Stages of the method for assessing threats to the economic security of the region from 219 its element.

In other words is oral language, not to be used in a scientific text… “In other words, it is necessary to select”

Of what? and use “assessing”: “Figure 4. - The playing field for evaluating the effectiveness.”

Do not start all captions by “the”: “Figure 5. - The playing field of an element of the region's economic security system: admittance-238 process-result.”

Conclusions section, as he abstract, should include

Brief contextualization and methodology, main findings and practical implications, thus justifying the importance and relevance of the text being presented and highlighting its originality and novelty…

Please, do not list, easy to write, but not the way to do it:

“The first category is characterized by a stable fall into the quartile of ineffective state 256 procurement. During the analyzed period of time, the competition indicators were below the average 257 for Russia. The average price drop also did not exceed the national average. This group includes 258 Dagestan, North Ossetia - Alania, Chechnya. 259

The second category is characterized by a stable positive situation, that is, over the course of five 260 years, competition and the fall in prices were above the average for Russia. These regions include 261 Altai Territory, Arkhangelsk Region, Astrakhan Region, Vladimir Region, Crimea, Kaluga Region, 262 Mordovia. 263

The third is unstable regions. Among them there are regions with a gradual improvement in 264 public procurement, for example, the Magadan region and the Moscow region, and one year of 265 worsening of the situation, for example, the Novgorod region, the Orenburg region.”

You may confirm that the expression “public procurement” is scarcely used in the text.

I could not find the supplementary material mentioned:

Supplementary Materials: The following are available online at www.mdpi.com/xxx/s1, Figure S1: title, Table 271 S1: title, Video S1: title.”

References, this is an international Journal, References must also include a range of international authors, to be relevant…

References must also be updated to include much more recent references.

Author Response

Thank your for your comments.

The file with highlighted changes is attached.

Important to mention thet the purpose of the study is to develop a method (Part 4). Approbation of it is shown in the framework of public procurement in the regions of Rissia, but it can be used wirh every element of economic system in every country.

Profreading from MDPI wil be made.

Round 2

Reviewer 2 Report

Authors mmade some effort to improve the paper. One important point wasn't done. The purpose of the study must be more specified in the text.

Author Response

Thank you for ypur comment.

The purpose of the study mhas been more specified in the introduction.

"From the authors’ point of view, it is important to identify threats that arise inside the system. For achieving this idea, it is important to understand the locations of the threats, that is, we need to know the elements of the system that can threaten it. Therefore, the purpose of this paper is to create a universal method for assessing threats to the economic security of the region. This method examines system elements that can potentially cause threats."

Reviewer 3 Report

Highlight changes in yellow in a next revision, please. No track changes.

Consider comments in the entire text.

Before, not after…

“Profreading from MDPI wil be made.”

I could not see any yellowed changes indicated in the text, to be able to focus on that…

Instead, it is in the “COVER LETTER” file… But no answer at all to the extensive comments.

Detailed answers, as it happen in every case, must be given to the reviewers…

MDPI instructions explain what to do.

This information in the system is NO answer:

“Author's Notes

Thank your for your comments.

The file with highlighted changes is attached.

Important to mention thet the purpose of the study is to develop a method (Part 4). Approbation of it is shown in the framework of public procurement in the regions of Rissia, but it can be used wirh every element of economic system in every country.”

Every review needs detailed answers, and I made extensive comments:

Highlight changes in yellow in a next revision, please. No track changes.

Consider comments in the entire text.

Revise title: “The” means there is no possible other…

Also, to start by the is not enlightening…

The title s important, reframe it:

 The method for assessing threats to the economic 2 security system of the region in the framework of 3 public procurement

See that the language is not clear and then it will reflect on the entire text...

Abstract…

What region? Economic security at what level or context?

“more topical”?

“Currently, the issue of economic security of the region is becoming more topical.”

A method, not the method… “ study is to develop a method”, compare with title…

Where? See that the title does not mention Russia… “regional economic”

I do not understand the sentence: “to the regional economic security 12 system from its elements.” What elements?!

Revise language, determined means something else:

“It was determined that the greatest risks for the system was in connection between competition and 23 budget savings.”

End with practical implications to highlight the importance of this text…

Keywords cannot be presented like this: “Economic security of the region;”

Maybe “Russia Economic security”

Revise the entire language mentioning region… in the complete manuscript…

Correct, have you double checked? “state procprocurement”

As seen, the English must be entirely proofread… It is not correct, just as example:

“In modern conditions of 36 digitalization progress can have as supporting influence to the system economic security of a region as 37 the potential threats”

This says nothing… Elements from 1 to n, no detail at all, see that Figures must be original and captions complete self-explanatory, which is not the case (the region “isse” again…):

Figure 1. – The system of economic security of the region.

Clarify…

“This figure schematically shows that the system of economic security of the region consists of a set of 59 interconnected elements with its own internal environment.”

After Introduction, results follow.

I acknowledge  authors are strictly following the template: https://www.mdpi.com/files/word-templates/risks-template.dot, here: https://www.mdpi.com/journal/risks/instructions.

Most of the Reviewers will be misled, because then the coherence is not there when authors state: “2. Results 65

The proposed method was tested on the assessment of threats to the economic security of the 66 region from public procurement by the following steps”

It is not a question of respecting the template, but also to adequate the text to the order indicated in the temple. The method was not yet presented…

Are theses results or methodology? See:

“Stage 1. In this study, public procurement was selected as an element of the region's economic 68 security system that can influence it.”

Authors are not clear, an international reader needs proper references for all similar cases…

“44 federal laws on the Tenderplan 74 analytical platform”

Revise all italics to variables: “Student's 102 t-test:”

Revise captions to be completely self-explanatory: “Table 1. – Results of regression analysis.”

Revise all headings in captions and add notes at the end of the tables, because they are available separately from the text:

y]x

”ed

There cannot be similar captions, not possible to different tables:

Table 2. - Results from regression analyses.”

And check English too

Again, italics, headings , notes, to be checked…

Clarify statements and hypothesis…

Which are presented later on? Methodology…

The structure of the text needs complete revision then…

“H1: The indicator percentage of applications requiring collateral to participate in the tender 114 affects the percentage of admissions.”

All equations must be properly introduced by number previously to being presented, as usual… and italics and units must be checked, after equations

Address the originality of equations to, no references means complete originality.

References are not presented after dots…

“but also the country as a whole. 152 (Chupaniva, 2016; Drozdov, 2019) 153

Fourthly, there is a co-directional”

Do not use the “listing” style: “Fourthly, there is”

One author then?! Possessive case… check English…

“Stage 7. The studied literature confirms the author's opinion”

What regions, what groups?

Table 3. - Distribution of regions by groups.”

Groups defined after the table? Add notes, there are much better ways to present nad clarity the data…

“Group 1 - ineffective state procurements. Regions with low competition, resulting in low price 174 reductions; 175

Group 2 - a threat to economic security. Regions with high levels of competition but suspiciously 176 low price reductions; 177

Group 3 - a threat to economic security. Regions with suspiciously high price levels with little 178 competition; 179

Group 4 - effective state procurements. Regions with a favorable atmosphere for the 180 development of the public procurement system, that is, a high level of high price reductions.”

Units are presented inside () and there must be headings to all:

Table 4. - Borders of quartiles for assessing the procurement activities of the regions of the Russian 186 Federation.”

Language must be revised…

3. Discussion 188

Consider the development trend of the regions of the Russian Federation in terms of the relationship 189 between competition and economy in the period from 2014 to 2018 (Table 5).”

And again, the discussion contains tables and figures, is it a discussion or results sections, either authors join or reframe…

A discussion should justify/support/debate the results and include references if necessary, which is not the case

All axis must have legends and units etc, check all, the first authors seems not to be used to the publish process but others are…

Not “1 group” but “group 1”, etc…

And again, the necessary clarification: “Figure 2. – Distribution of regions by groups, 2014-2018.”

 Believe authors must dedicate a significant further effort in revising the entire language, namely considering the followed structure in the manuscript…

4. Materials and Methods 213

In general, a method for assessing threats to the economic security of a region from its elements 214 can be a sequence of interrelated actions leading to ranking regions regarding threats and the 215 resulting measures to counter their spread. These measures are supposed to be considered by the 216 state authorities. The method is shown in the Figure 3.”

Then a method or the method again… or “region”…

Figure 3 – Stages of the method for assessing threats to the economic security of the region from 219 its element.

In other words is oral language, not to be used in a scientific text… “In other words, it is necessary to select”

Of what? and use “assessing”: “Figure 4. - The playing field for evaluating the effectiveness.”

Do not start all captions by “the”: “Figure 5. - The playing field of an element of the region's economic security system: admittance-238 process-result.”

Conclusions section, as he abstract, should include

Brief contextualization and methodology, main findings and practical implications, thus justifying the importance and relevance of the text being presented and highlighting its originality and novelty…

Please, do not list, easy to write, but not the way to do it:

“The first category is characterized by a stable fall into the quartile of ineffective state 256 procurement. During the analyzed period of time, the competition indicators were below the average 257 for Russia. The average price drop also did not exceed the national average. This group includes 258 Dagestan, North Ossetia - Alania, Chechnya. 259

The second category is characterized by a stable positive situation, that is, over the course of five 260 years, competition and the fall in prices were above the average for Russia. These regions include 261 Altai Territory, Arkhangelsk Region, Astrakhan Region, Vladimir Region, Crimea, Kaluga Region, 262 Mordovia. 263

The third is unstable regions. Among them there are regions with a gradual improvement in 264 public procurement, for example, the Magadan region and the Moscow region, and one year of 265 worsening of the situation, for example, the Novgorod region, the Orenburg region.”

You may confirm that the expression “public procurement” is scarcely used in the text.

I could not find the supplementary material mentioned:

Supplementary Materials: The following are available online at www.mdpi.com/xxx/s1, Figure S1: title, Table 271 S1: title, Video S1: title.”

References, this is an international Journal, References must also include a range of international authors, to be relevant…

References must also be updated to include much more recent references. ”

(…)

Title: The English is not OK: “ A method for the assessing threats to the economic 2 security of the region

Again, to be clarified:

“ The purpose 11 of this study is to develop a method for assessing threats to the economic security of the region.”

The English is incorrect. This sentence has no conclusion: “Based on these groups might develop individual economic 27 security monitoring tools in each region.”

Theses are non-comprehensible sentences: “ In modern conditions of digitalization, progress can have the same supportive 38 impact on the systemic economic security of the region as potential threats to it”

What Figure, specify: “This figure schematically shows that the system of economic security of the region consists of a set of 61 interconnected elements.”

Unclear content… “These elements with its own internal environment are from economic, social, 62 ecological or other spheres.”

See that italics were not applied to parameters, mathematical nomenclature… all over

“Table 2. – Main indicators of regression analysis. 119 #    Model SE       β          p-Value           R-squared”

Wite sentences differently:

“The model of H1 is represented in the equation 1:”

“The model of H2 is represented in the equation 2:”

Etc…

Authors opted to ignore proposed changes and did not answer

Changes were mostly cosmetic.

The language needs strong revision, it is not possible to properly review a text in poor English

Please careful check previous comments and enlighten me with clear answers.

Author Response

Proofreading has been done.

‘Revise title: “The” means there is no possible other…

Also, to start by the is not enlightening…

The title s important, reframe it:

“ The method for assessing threats to the economic security system of the region in the framework of public procurement

 See that the language is not clear and then it will reflect on the entire text...’ – The title has been changed to “A method for assessing threats to the economic security of a region: a case study of public procurement in Russia”

Abstract…

‘What region? Economic security at what level or context? – “more topical”? – no more“Currently, the issue of economic security of the region is becoming more topical.”A method, not the method… “ study is to develop a method”, compare with title…Where? See that the title does not mention Russia… “regional economic”I do not understand the sentence: “to the regional economic security 12 system from its elements.” What elements?!’

 Every region (province, territory inside the country because here we consider the economic security on meso-level.

“End with practical implications to highlight the importance of this text…” – The importance of this method is that with the help of grouping regions (territory of the country) based on proposed method it is possible to develop individual economic security monitoring tools. As a result the efficiency of the region can be higher.

‘Keywords cannot be presented like this: “Economic security of the region;”Maybe “Russia Economic security”’ – we add key word “regions of Russia”, but this paper is for every economic security system and economic security of its element.

“Revise the entire language mentioning region… in the complete manuscript…” – region is in the meaning of territory, part of the country

Correct, have you double checked? “state procprocurement” – you are right. Public procurement.

‘As seen, the English must be entirely proofread… It is not correct, just as example: “In modern conditions of 36 digitalization progress can have as supporting influence to the system economic security of a region as 37 the potential threats” This says nothing… Elements from 1 to n, no detail at all, see that Figures must be original and captions complete self-explanatory, which is not the case (the region “isse” again…):’ – proofreading is done.

‘“Figure 1. – The system of economic security of the region.

 Clarify…

“This figure schematically shows that the system of economic security of the region consists of a set of 59 interconnected elements with its own internal environment.”’ - Figure 1 schematically shows that the system of economic security of the region consists of a set of interconnected elements. Each element has regulatory and legal acts that govern, subjects of management, objects of management, possibility of negative impacts and consequences of their impact on the system, a group of indicators and indicators that characterise the system (Feofilova, 2014). In the system, the elements interact . Its environments both positively and negatively affect the general environment of economic security of the region.

After Introduction, results follow.

‘I acknowledge  authors are strictly following the template: https://www.mdpi.com/files/word-templates/risks-template.dot, here: https://www.mdpi.com/journal/risks/instructions. Most of the Reviewers will be misled, because then the coherence is not there when authors state: “2. Results 65 The proposed method was tested on the assessment of threats to the economic security of the 66 region from public procurement by the following steps” It is not a question of respecting the template, but also to adequate the text to the order indicated in the temple. The method was not yet presented…’ - This work is devoted to public procurement, which is one of the most important elements of the system of economic security of the region (Grandia and Kruyen, 2020). The approbation of the developed method is assessing threats to the economic security of the Russian regions in the framework of public procurement. The stages and details of the method are presented in the 4th part of this article, Materials and Method.

Are theses results or methodology? See: “Stage 1. In this study, public procurement was selected as an element of the region's economic 68 security system that can influence it.” – it is the result of step by step used of method.

Authors are not clear, an international reader needs proper references for all similar cases…

“44 federal laws on the Tenderplan 74 analytical platform” – we’ve escaped this. Public procurement will be enough for the readers.

Revise captions to be completely self-explanatory: “Table 1.  Results of regression analysis.” - Table 1. Results of regression analysis: direction of changes in related indicators

Revise all headings in captions and add notes at the end of the tables, because they are available separately from the text:

y]x

”ed

There cannot be similar captions, not possible to different tables:

Table 2. - Results from regression analyses.” - Table 2. Main indicators of regression analysis.

And check English too

Again, italics, headings , notes, to be checked…

Clarify statements and hypothesis…

Which are presented later on? Methodology…

The structure of the text needs complete revision then…

“H1: The indicator percentage of applications requiring collateral to participate in the tender 114 affects the percentage of admissions.”

All equations must be properly introduced by number previously to being presented, as usual… and italics and units must be checked, after equations

Address the originality of equations to, no references means complete originality.

References are not presented after dots…

“but also the country as a whole. 152 (Chupaniva, 2016; Drozdov, 2019) 153

Fourthly, there is a co-directional”

Do not use the “listing” style: “Fourthly, there is”

One author then?! Possessive case… check English…

“Stage 7. The studied literature confirms the author's opinion”

“What regions, what groups? “Table 3. - Distribution of regions by groups.” Groups defined after the table? Add notes, there are much better ways to present nad clarity the data… “Group 1 - ineffective state procurements. Regions with low competition, resulting in low price 174 reductions; 175 Group 2 - a threat to economic security. Regions with high levels of competition but suspiciously 176 low price reductions; 177 Group 3 - a threat to economic security. Regions with suspiciously high price levels with little 178 competition; 179 Group 4 - effective state procurements. Regions with a favorable atmosphere for the 180 development of the public procurement system, that is, a high level of high price reductions.””- We’ve changed the place and clarified the meaning of the groups. Groups are based on the connection of number of bidders and the average price drop. So the can be effective or ineffective and threating to economic security (2 ways). We have grouped them just under this condition.

Units are presented inside () and there must be headings to all:

Table 4. - Borders of quartiles for assessing the procurement activities of the regions of the Russian 186 Federation.”

Language must be revised…

3. Discussion 188

‘Consider the development trend of the regions of the Russian Federation in terms of the relationship 189 between competition and economy in the period from 2014 to 2018 (Table 5).” And again, the discussion contains tables and figures, is it a discussion or results sections, either authors join or reframe… A discussion should justify/support/debate the results and include references if necessary, which is not the case’ – it is important to use these tables and figures here because it is the beginning of discussion. It means that this grouping is results of some problems, which are discussed here.

All axis must have legends and units etc, check all, the first authors seems not to be used to the publish process but others are… - in the process

Not “1 group” but “group 1”, etc…

And again, the necessary clarification: “Figure 2. – Distribution of regions by groups, 2014-2018.”

 Believe authors must dedicate a significant further effort in revising the entire language, namely considering the followed structure in the manuscript…

4. Materials and Methods 213

In general, a method for assessing threats to the economic security of a region from its elements 214 can be a sequence of interrelated actions leading to ranking regions regarding threats and the 215 resulting measures to counter their spread. These measures are supposed to be considered by the 216 state authorities. The method is shown in the Figure 3.”

Then a method or the method again… or “region”…

Figure 3 – Stages of the method for assessing threats to the economic security of the region from 219 its element.

In other words is oral language, not to be used in a scientific text… “In other words, it is necessary to select” - changed

Of what? and use “assessing”: “Figure 4. - The playing field for evaluating the effectiveness.” - Figure 4. Playing field for evaluating the effectiveness of the public sector.

Do not start all captions by “the”: “Figure 5. - The playing field of an element of the region's economic security system: admittance-238 process-result.” - Figure 5. Playing field for evaluating the effectiveness of an element of the region's economic security system: admittance–process–result.

Conclusions section, as he abstract, should include

Brief contextualization and methodology, main findings and practical implications, thus justifying the importance and relevance of the text being presented and highlighting its originality and novelty… - A method for assessing threats to the system of economic security of the region from system’s elements  was developed. The method consists of eight sequential stages, which can be applied to every element of the system of economic security of the region or territory of every country.

Approbation of the method  in the framework of public procurement in Russian regions revealed that the non-standard ratio of competition indicators to economy indicators entails a threat to the economic security of the region. Based on this connection, four groups of regions were proposed. Regions of the RF were separated into groups. From the analysis of the dynamics of the regions of the RF, the groups were divided into several categories.

Please, do not list, easy to write, but not the way to do it:

“The first category is characterized by a stable fall into the quartile of ineffective state 256 procurement. During the analyzed period of time, the competition indicators were below the average 257 for Russia. The average price drop also did not exceed the national average. This group includes 258 Dagestan, North Ossetia - Alania, Chechnya. 259

The second category is characterized by a stable positive situation, that is, over the course of five 260 years, competition and the fall in prices were above the average for Russia. These regions include 261 Altai Territory, Arkhangelsk Region, Astrakhan Region, Vladimir Region, Crimea, Kaluga Region, 262 Mordovia. 263

The third is unstable regions. Among them there are regions with a gradual improvement in 264 public procurement, for example, the Magadan region and the Moscow region, and one year of 265 worsening of the situation, for example, the Novgorod region, the Orenburg region.”

I could not find the supplementary material mentioned:

Supplementary Materials: The following are available online at www.mdpi.com/xxx/s1, Figure S1: title, Table 271 S1: title, Video S1: title.” – no supplementary materials

References, this is an international Journal, References must also include a range of international authors, to be relevant…

References must also be updated to include much more recent references. ” – we have added international authors, By the way practically all references are from international journals … due to this it should rather interesting topic for all.

Round 3

Reviewer 3 Report

Highlight changes in yellow in a next revision, please. No track changes.

Consider comments in the entire text.

No clear answers were given, but comments and added text, it is not the way to answer to the reviewrs…

Clarify the title… “

 A method for assessing threats to the economic 2 security of a region” says nothing…

Language must be enlightening, to what kind of models do you refer to?!

“ As a result, four significant models were identified”

I do not understand…

The answer relates to content tat is not present on the text!!

““End with practical implications to highlight the importance of this text…” – The importance of this method is that with the help of grouping regions (territory of the country) based on proposed method it is possible to develop individual economic security monitoring tools. As a result the efficiency of the region can be higher.”

Captions are not comments, but legends, it must be entirely correct, move part to the text…

“Figure 1 schematically shows that the system of economic security of the region consists of a set of 60 interconnected elements. Each element has regulatory and legal acts that govern, subjects of 61 management, objects of management, possibility of negative impacts and consequences of their impact 62 on the system, a group of indicators and indicators that characterise the system (Feofilova, 2014). In the 63 system, the elements interact. Its environments both positively and negatively affect the general 64 environment of economic security of the region. Simultaneously, the system of economic security, 65 which is exposed to external factors, has both a positive impact on its elements and a negative impact 66 on its elements.”

Revise language:

“This work is devoted to public procurement,”

I am sorry but Table 1 is not clear, neither caption nor content, headings are necessary…

Table 1. Results of regression analysis: direction of changes in related indicators”

Revise italics to parameters in Table 2, only as example…

SE

β

p-Value

R-squared

And add notes to define…”

Extensicve comments and no answer…

“And check English too

Again, italics, headings , notes, to be checked…

Clarify statements and hypothesis…

Which are presented later on? Methodology…

The structure of the text needs complete revision then…

“H1: The indicator percentage of applications requiring collateral to participate in the tender 114 affects the percentage of admissions.”

All equations must be properly introduced by number previously to being presented, as usual… and italics and units must be checked, after equations

Address the originality of equations to, no references means complete originality.

References are not presented after dots…

“but also the country as a whole. 152 (Chupaniva, 2016; Drozdov, 2019) 153

Fourthly, there is a co-directional”

Do not use the “listing” style: “Fourthly, there is”

One author then?! Possessive case… check English…

“Stage 7. The studied literature confirms the author's opinion”

And many more

Revise all italics to variables, it has to do with mathematical nomenclature, as usual…!

See that a reader focusing on a separate section will never understand to what kind of method are authors referring to…

Clarify language.

5. Conclusions 264

A method for assessing threats to the system of economic security of the region from system’s 265 elements was developed. The method consists of eight sequential stages, which can be applied to 266 every element of the system of economic security of the region or territory of every country.”

(…)

“Based on the proposed method, a tool for monitoring the degree of economic security in the 283 region for the field of public procurement can be created, which enables the tracking of a potential 284 threat to the entire system of economic security during tendering procedures using digital 285 technologies. This method can”

Yellow them:

References, this is an international Journal, References must also include a range of international authors, to be relevant…

References must also be updated to include much more recent references. ” – we have added international authors, By the way practically all references are from international journals … due to this it should rather interesting topic for all.

Please revise the text considering all the comments until now, focusing in language

Author Response

thank you for ypur comments. you have answered to them and made the replace in the paper.

Clarify the title… “ A method for assessing threats to the economic security of a region” says nothing… - In the title authors wanted to show that they have developed new method that can assess threats to the economic security of the region (in the meaning of the territory).It is necessary to assess the possibility of a threat, that is, hypothetically, in this environment, adverse consequences may occur.

Language must be enlightening, to what kind of models do you refer to?! – We refer to the models of pairwise regression analysis. We have changed the sentence “ As a result, four significant models were identified” to “As a result of pairwise regression models analysis we have defined four significant relationships between public procurement indicators.”

The answer relates to content tat is not present on the text!! – it is practical use of this method. We can show it on the example of Russian regions, but it can be hard to understand for international readers. That is why we just say that this method should be used in thi way.

““End with practical implications to highlight the importance of this text…” – The importance of this method is that with the help of grouping regions (territory of the country) based on proposed method it is possible to develop individual economic security monitoring tools. As a result the efficiency of the region can be higher.”

Captions are not comments, but legends, it must be entirely correct, move part to the text… - We have decided to add legends inside the figure 1. It should be easier to understand the meaning of the picture.

“Figure 1 schematically shows that the system of economic security of the region consists of a set of 60 interconnected elements. Each element has regulatory and legal acts that govern, subjects of 61 management, objects of management, possibility of negative impacts and consequences of their impact 62 on the system, a group of indicators and indicators that characterise the system (Feofilova, 2014). In the 63 system, the elements interact. Its environments both positively and negatively affect the general 64 environment of economic security of the region. Simultaneously, the system of economic security, 65 which is exposed to external factors, has both a positive impact on its elements and a negative impact 66 on its elements.”

Revise language:

“This work is devoted to public procurement,” – this work focuses on public procurement ….

 I am sorry but Table 1 is not clear, neither caption nor content, headings are necessary… - Before the Table 1 we have described every sign. For better understanding we have added the numbering of indicators several times. “The summary table (Table 1) shows the direction of the changes in the related indicators (1-12), provided that the indicator is significant. The significance of the indicator was determined based on the Student's t-test: (“↑↑”) - indicators have a co-directional change, that is, with an increase (decrease) in the influencing indicator, the dependent variable increases (decreases); (“↑↓”) - indicators have a multidirectional change, that is, with an increase (decrease) in the influencing indicator, the dependent variable decreases (increases). If the value of the t-criterion is greater than the significance level, then the null hypothesis about the insignificance of the regression coefficient is accepted, that is, there is no connection among the analysed indicators "-".”

Table 1. Results of regression analysis: direction of changes in related indicators”

Revise italics to parameters in Table 2, only as example… - there is no italics in Table 2. we have added the explanation of the [parameters behind the table. “The column Model shows the number of dependent and independent variable. β-coefficient shows how much we expect dependent variable to change by given a change in independent variable. R-square shows the amount of variance of dependent variable explained by independent variable. P-value is used to show that each coefficient is different from 0”

“SE

β

p-Value

R-squared

 And add notes to define…”

Clarify statements and hypothesis… - The idea of hypotheses is represented in the Stage 4. “Stage 4. Based on the groups obtained, their influence on each other was revealed: the admittance group affects the process and the results, and the process group affects the results group. Hypotheses were formed to test them.”. Due to your comments we have decided that it is better to take away the formed hypotheses from the text and stay just equations which are completely original as they are based on this research. They are just results of regression modeling ()

Which are presented later on? Methodology…

 The structure of the text needs complete revision then…

“H1: The indicator percentage of applications requiring collateral to participate in the tender 114 affects the percentage of admissions.”

 All equations must be properly introduced by number previously to being presented, as usual… and italics and units must be checked, after equations

Address the originality of equations to, no references means complete originality.

References are not presented after dots…

“but also the country as a whole. 152 (Chupaniva, 2016; Drozdov, 2019) 153 – references are before the dot

Fourthly, there is a co-directional”

 Do not use the “listing” style: “Fourthly, there is” – replace with Moreover. It is important to explain several points about the getting connections.

One author then?! Possessive case… check English…

“Stage 7. The studied literature confirms the author's opinion”- Stage 7. The literature confirms the authors' opinions

See that a reader focusing on a separate section will never understand to what kind of method are authors referring to… - I am sorry but we can’t explain the method 2 times. If the reader doesn’t get the idea he can look through the section about method …

Clarify language.

5. Conclusions 264

A method for assessing threats to the system of economic security of the region from system’s 265 elements was developed. The method consists of eight sequential stages, which can be applied to 266 every element of the system of economic security of the region or territory of every country.”

 (…)

 “Based on the proposed method, a tool for monitoring the degree of economic security in the 283 region for the field of public procurement can be created, which enables the tracking of a potential 284 threat to the entire system of economic security during tendering procedures using digital 285 technologies. This method can”

 The purpose is to create the method. We have done it. For the question why have we done it? what reason? We can say that with its help special tools for monitoring can be created. It is just the possibility … with the help of this method as it shows where it is important to pay more attention.

Yellow them:

References, this is an international Journal, References must also include a range of international authors, to be relevant… - more than the half of the references has been found in Scopus …

References must also be updated to include much more recent references. ” – we have added international authors, By the way practically all references are from international journals … due to this it should rather interesting topic for all.

Please revise the text considering all the comments until now, focusing in language

Round 4

Reviewer 3 Report

Highlight changes in yellow in a next revision, please. No track changes.

Consider comments in the entire text.

In my perspective, the authors should improve the title

Again: “Clarify the title… “

 A method for assessing threats to the economic 2 security of a region” says nothing…”

The language has not been improved. It is not clear nor enlightening…

Abstract: The issue of economic security is becoming increasingly urgent. The purpose of this article 11 is to develop a method for assessing threats to the economic security of the region.”

(…)

(…)

See that this equation, only as example, needs italics (and remove the comma):

“y=45.77+0.56x, ”

These are not mere details…

Units are presented inside (): “Average price drop, % ”

Not “1 group” but “Group 1”: Table 5. Number of regions in the group, 2014–2018.

The caption does not allow the reader to know whats is excalty being presented…

Conclusions must start with a brief contextuaçlization to defend the need to publish this text…

Then… methodology

Contentes are mixed, see…

5. Conclusions 266

A method for assessing threats to the system of economic security of the region from system’s 267 elements was developed. The method consists of eight sequential stages, which can be applied to 268 every element of the system of economic security of the region or territory of every country. 269

Approbation of the method in the framework of public procurement in Russian regions revealed 270 that the non-standard ratio of competition indicators to economy indicators entails a threat to the 271 economic security of the region. Based on this connection, four groups of regions were proposed. 272 Regions of the RF were separated into groups. From the analysis of the dynamics of the regions of 273 the RF, the groups were divided into several categories.”

I had read all the answers given, which were had to find, not separated from comments, they are not satisfactory to me.

I have said all I wanted before.

I will not repeat now.

The changes made by the authors are mostly cosmetics.

Due to the language used in the text, the text is just not relevant.

It needed to be revised in order to be improved

Even if content is explained in the text, a reader focusing in a Figure or table must me completely enlightened, see that MDPI offers to reader the possibility of focusing in Figures and tables alone, so the notes are absolutely necessary

References are scarce.

Despite focusing in Russia, this is an international journal and international authors should be cited to aim a broader perspective.

Author Response

Dear Reviewer,
Thank you for your reply and useful recommendations. We have thoroughly revised the manuscript in compliance with the recommendations of referees. 
We have coreected the style of the text and use the order of the text which is on the site https://www.mdpi.com/journal/risks/instructions.

We added more imformation about used method in the abstract. And practical recommendations that can be used in Russia in the part results. All changes are highlighted with the yellow colour.

We hope that you will find the manuscript in its present form suitable for publication.
Best regards